# SEGMENTING THE UNKNOWN: DISCRETE DIFFUSION MODELS FOR NON-DETERMINISTIC SEGMENTATION

## ABSTRACT

Safety critical applications of deep-learning require models able to handle ambiguity and uncertainty. We introduce discrete diffusion models to capture uncertainty in semantic segmentation, with application in both oncology and autonomous driving. Unlike prior approaches that tackle these tasks in distinct ways, we formulate both as estimating a complex posterior distribution over images, and present a unified solution that leverages the discrete diffusion framework. Our contributions include the adaptation of discrete diffusion for semantic segmentation to model uncertainty and the introduction of an auto-regressive diffusion framework for future forecasting. Experimental evaluation on medical imaging data and real-world future prediction tasks demonstrates the superiority of our generative framework over deterministic models and its competitive performance compared to methods specific to these domains separately.

## 1 INTRODUCTION

Several real world applications require developing computer vision models capable of handling ambiguity and uncertainty. For example, autonomous driving applications need to model the decision-making processes of drivers and pedestrians to predict their positions in the future. Another example is the case of medical imaging where physicians oftentimes do not agree on a single diagnosis, hence we desire models capable of providing these multiple plausible predictions. However, the majority of models are still trained in a deterministic manner, meaning they provide only one prediction for each input. As a result, they can provide only one possible output for a given input, regardless of inherent uncertainty. Moreover, these models often exhibit high confidence in their predictions even when they are incorrect.

In this work, we focus on the task of semantic segmentation and investigate the use of discrete diffusion models (Austin et al., 2021) for handling ambiguous predictions. In particular, we tackle two non-deterministic semantic segmentation tasks: future prediction and medical image segmentation. Existing works in medical image segmentation (Kohl et al., 2019) utilize generative models such as VAEs to model the uncertainty of the prediction. On the other hand, although future segmentation is also inherently ambiguous, existing works (Lin et al., 2021) approach it as a deterministic task. Conversely, we treat both problems in a unified way, namely as a conditional generative task. Specifically, for the case of medical image segmentation we generate a segmentation mask given the input image, whereas for future segmentation we generate the next segmentation mask given the previous ones. Since diffusion models have demonstrated great capabilities in efficiently generating highly detailed and high resolution samples (Rombach et al., 2022), we opt to use them as our generative model. In particular, we utilize the discrete diffusion framework of Austin et al. (2021), which dovetails perfectly with the discrete nature of the predicted segmentation masks.

In summary, we make the following contributions: We are the first to model uncertainty of predictions using discrete diffusion models in semantic segmentation. Moreover for the case of future forecasting, we introduce an auto-regressive diffusion framework, that utilizes segmentation generations as inputs to predict further into the future. We evaluate our model on a Lung Cancer medical Imaging Dataset (LIDC) (Armato III et al., 2011) and on two future prediction tasks. The first is a car intersection simulator where we want to predict the path of the cars in the intersections, and the second is the future segmentation task in the Cityscapes dataset (Cordts et al., 2016). From our experimental evaluation, we show that our model consistently outperforms an equivalent deterministic model in

all tasks. Moreover, for the two real-world tasks, we demonstrate that our generative framework outperforms existing VAE-based methods on LIDC while performing on par with the state-of-the-art on Cityscapes future prediction.

## 2 RELATED WORK

**Non-deterministic segmentation** As precursors of ambiguous segmentation, Kohl et al. (2018) proposed the Probabilistic U-Net and later its hierarchical version (Kohl et al., 2019). It sparked further works from Baumgartner et al. (2019); Bhat et al. (2022), among others. These works are all centered on the use of variants of a conditional variational auto-encoder to model the ambiguity. In contrast, we propose to use diffusion models, similarly to the recent work by Rahman et al. (2023). We differ in that we use discrete diffusion, and extend the study to future segmentation prediction.

**Diffusion and Discrete Diffusion** Our approach towards handling ambiguity is to use generative models and be able to sample possible outcomes. In this work, we investigate Diffusion Probabilistic Models (DPM), which are generative models first formalised by Sohl-Dickstein et al. (2015) and later successfully applied to high quality image synthesis by Ho et al. (2020). We specifically base our work on Discrete Denoising Diffusion Probabilistic Models (D3PMs) introduced by Austin et al. (2021). We found D3PMs to offer substantial flexibility regarding the modeling of the corruption process, and to be particularly well suited to the discrete nature of the segmentation task.

**Diffusion in Segmentation** Since diffusion gained strong popularity, numerous works have tried applying DPMs to segmentation. Most of them used it in the context of medical image segmentation (Wu et al., 2022; 2023; Wolleb et al., 2022b;a; Pinaya et al., 2022), but contrary to us none of these aimed to solve the inherent ambiguity. Only the previously mentioned Rahman et al. (2023) is leveraging diffusion for handling ambiguity of medical images.

Few works have applied it to other domains: Amit et al. (2021) applies DPMs to the interactive segmentation task, which is a restricted two classes problem. Chen et al. (2022a) uses another form of discrete DPMs (Chen et al., 2022b) to perform panoptic segmentation of full images and videos. Similarly to us, they introduce an auto-regressive scheme, but do not tackle any ambiguous tasks

**Future segmentation** On the future segmentation forecasting task, different approaches exist. Direct semantic forecasting introduced by Luc et al. (2018) directly predicts the future segmentation map from past ones (Bhattacharyya et al., 2018; Rochan et al., 2018; Chen & Han, 2019). In this study, we follow this methodology for the future prediction tasks, primarily for the sake of simplicity. In contrast, feature level forecasting predicts future intermediate features from past ones as done by Luc et al. (2018); Saric et al. (2020); Chiu et al. (2020) and (Lin et al., 2021) which is the best performing method in this task.

## 3 METHOD

### 3.1 DIFFUSION MODELS

Given samples $\boldsymbol{x}_0 \sim q(\boldsymbol{x}_0)$ from a data distribution, diffusion models (Sohl-Dickstein et al., 2015; Ho et al., 2020) are characterized by a forward Markov process $q(\boldsymbol{x}_{1:T}|\boldsymbol{x}_0)$ - that destroys the data $\boldsymbol{x}_0$ through the successive addition of noise creating a sequence of increasingly corrupted latent variables $\boldsymbol{x}_1, \ldots, \boldsymbol{x}_T$ - and by the corresponding reverse process $p_\theta(\boldsymbol{x}_{0:T})$ which is trained to gradually remove the noise. For the case of continuous data, the forward process usually adds Gaussian noise, whereas for discrete data any Markov transition matrix can be used as a corruption.

During training, the goal is to learn the reverse process such that at inference the model can generate data by gradually denoising a random input. The usual diffusion optimization process consists of minimizing a variational upper bound of the negative log-likelihood as follows

$$
\begin{aligned}
L_{\text{vb}} = \mathbb{E}_{q(\boldsymbol{x}_0)} \bigg[ & D_{\text{KL}} \left[ q\left(\boldsymbol{x}_T \mid \boldsymbol{x}_0\right) \| p\left(\boldsymbol{x}_T\right)\right] + \sum_{t=2}^{T} \mathbb{E}_{q(\boldsymbol{x}_t|\boldsymbol{x}_0)} \left[ D_{\text{KL}} \left[ q\left(\boldsymbol{x}_{t-1} \mid \boldsymbol{x}_t, \boldsymbol{x}_0\right) \| p_\theta\left(\boldsymbol{x}_{t-1} \mid \boldsymbol{x}_t\right)\right]\right] \\
& - \mathbb{E}_{q(\boldsymbol{x}_1|\boldsymbol{x}_0)} \left[\log p_\theta\left(\boldsymbol{x}_0 \mid \boldsymbol{x}_1\right)\right] \bigg].
\end{aligned}
$$

(1)

## 3.2 DISCRETE DIFFUSION

As previously mentioned, for the case of discrete random variables, where $\boldsymbol{x}_0, \ldots, \boldsymbol{x}_T$ are one-hot vectors, arbitrary Markov transition matrices can be employed to define the forward process. In particular, following Austin et al. (2021), at the $t$-th forward step a transition probability matrix $\boldsymbol{Q}_t$ is used such that: $q(\boldsymbol{x}_t \mid \boldsymbol{x}_{t-1}) = \boldsymbol{x}_{t-1}\boldsymbol{Q}_t$. Note that the transitions are applied independently to each input dimension (e.g. pixel).

In order to use the matrices $\boldsymbol{Q}_t$ for a discrete diffusion model they need to have the following two properties. First, we want $\overline{\boldsymbol{Q}}_T = \boldsymbol{Q}_1\boldsymbol{Q}_2 \ldots \boldsymbol{Q}_T$ to converge to a stationary distribution in order to be able to sample $\boldsymbol{x}_T$ to start the denoising process. Second, we want to be able to efficiently compute $\overline{\boldsymbol{Q}}_t$ which enables efficient sampling of $\boldsymbol{x}_t$ from the $t$-step marginal $q(\boldsymbol{x}_t|\boldsymbol{x}_0) = \boldsymbol{x}_0\overline{\boldsymbol{Q}}_t$ for any $t$. This permits efficient training as we can optimize eq. (1) one term at a time with SGD without having to compute the full forward process for every gradient step. While Austin et al. (2021) proposes various alternatives for $\boldsymbol{Q}_t$ that satisfy these conditions, in this work, we found uniform transition matrices to work best, ie. matrices with uniform transition probabilities written as $\boldsymbol{Q}_t = (1 - \beta_t)\mathbf{I} + \beta_t \mathbf{11}^T/K$, where $\mathbf{1}$ is a column vector of all ones and $K$ the number of possible states.

To learn the reverse process, we train a model to predict $\boldsymbol{x}_0$ given $\boldsymbol{x}_t$, denoted as $\widetilde{p}_\theta(\widetilde{\boldsymbol{x}}_0 \mid \boldsymbol{x}_t)$, where $\theta$ are the model parameters. Subsequently, to get the transition probabilities $p_\theta(\boldsymbol{x}_{t-1} \mid \boldsymbol{x}_t)$, we marginalise over $\boldsymbol{x}_0$ as follows:

$$p_\theta(\boldsymbol{x}_{t-1} \mid \boldsymbol{x}_t) = \sum_{\widetilde{\boldsymbol{x}}_0} q(\boldsymbol{x}_{t-1} \mid \boldsymbol{x}_t, \widetilde{\boldsymbol{x}}_0)\, \widetilde{p}_\theta(\widetilde{\boldsymbol{x}}_0 \mid \boldsymbol{x}_t).$$

where the posterior $q(\boldsymbol{x}_{t-1} \mid \boldsymbol{x}_t, \boldsymbol{x}_0)$ has a simple analytical expression that can be derived using Bayes' rule and the Markov property.

Finally, we utilize a prediction loss $L_p = \mathbb{E}_{q(\boldsymbol{x}_0)}\mathbb{E}_{q(\boldsymbol{x}_t|\boldsymbol{x}_0)}[-\log \widetilde{p}_\theta(\boldsymbol{x}_0 \mid \boldsymbol{x}_t)]$ which encourages the correct prediction of $\boldsymbol{x}_0$ at each step of the reverse process. Overall, we optimize $L = L_{\mathrm{vb}} + \lambda L_p$ as a combination of $L_{\mathrm{vb}}$ with this prediction loss $L_p$ scaled by a scalar $\lambda$.

## 3.3 DISCRETE DIFFUSION MODELS FOR SEGMENTATION

In the context of semantic segmentation, every random variable $\boldsymbol{x}$ corresponds to one pixel in the image. The number of different states $K$ corresponds to the number of segmented classes in the dataset, which is generally a low number below 100, so we do not encounter scaling-related problems.

We adopt the Unet architecture classically used in diffusion, and in particular we adapt code from Wang (2023). The default Image Diffusion Unet architecture takes as input an image with the current time step, and returns a slightly less noisy version of that image. For the Segmentation Diffusion Unet, we instead have a corrupted segmentation map as input and get a denoised version of it as output, ie. a prediction $\tilde{\boldsymbol{x}}_0$ of the clean segmentation. Since segmentation maps are discrete, we first project them into a continuous vector space of dimension $E$ with an embedding layer, and convert the Unet output to the probability distribution $\widetilde{p}_\theta(\widetilde{\boldsymbol{x}}_0 \mid \boldsymbol{x}_t)$ with a softmax layer. This distribution is then used to compute $p_\theta(\boldsymbol{x}_{t-1} \mid \boldsymbol{x}_t)$ and sample from it.

**Input Conditioning**   Our aim is to generate the segmentation map that corresponds to the input frame, not just any random one. Consequently, we must condition the generation process accordingly. To do so, we concatenate the conditioning tensor channel-wise to the embedded noisy segmentation, and use this as input to the Unet. Among the different conditioning approaches we tried, we found the simple concatenation to be fast and perform consistently well across all tasks.

In this work, we tackle tumor segmentation and future prediction, for which we use slightly different conditioning tensors. For the medical segmentation task, we directly use the one-channel gray-scale lung scan as conditioning input. For the future segmentation task, we instead use the concatenation of multiple previous segmentations as conditioning. These segmentations are evenly spaced in time and are embedded in the same space as the noisy input $\boldsymbol{x}_t$. The rationale for this choice is clarified below.

**Time Auto-Regressive Diffusion**   Conditioning with past segmentation masks, as described above, allows us to forecast the segmentation of a frame occurring at a specific time interval from the current one. To extend our predictive horizon, we propose auto-regressively predicting the next segmentation,

which involves using the predicted segmentation masks as new inputs to the model. The overall model architectures used are presented in fig. 1. Using our Segmentation Diffusion model auto-regressively constraints the kind of inputs our model can use because the inputs and outputs must be of the same type ie. segmentation masks. As a result, we rely only on previous segmentations as conditioning inputs for our future segmentation networks.

However, we point out that restricting the input exclusively to segmentation masks limits the information available to the model to make its predictions. To illustrate this with a concrete example, consider the scenario where a group of three pedestrians may be represented as a single, undifferentiated blob on the segmentation map. This choice can clearly hinder the model's ability to discern individual objects or finer details within the scene, thereby affecting the accuracy of its predictions.

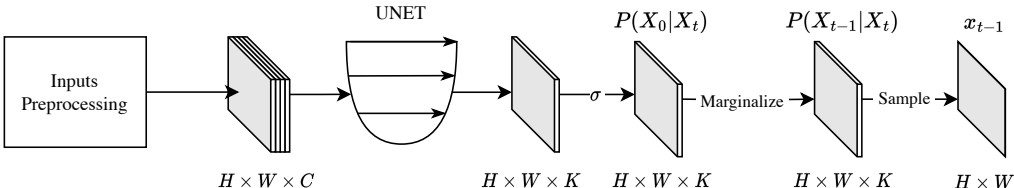

(a) Overall architecture of Discrete Diffusion Models used for segmentation.

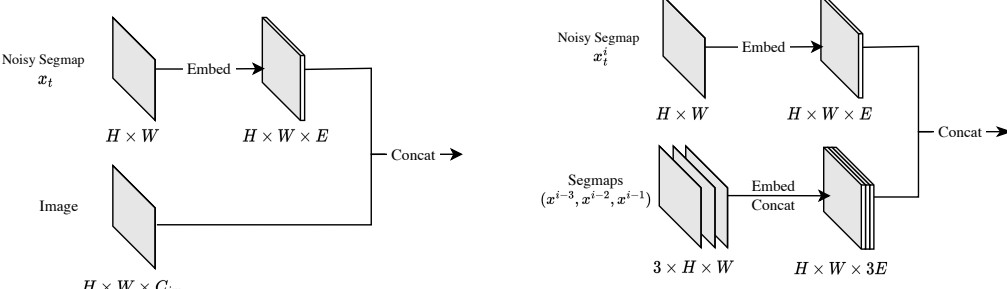

(b) Input preprocessing block used for the medical segmentation task. Conditioning inputs are one-channel images representing lung scans.

(c) Input preprocessing block used for the future prediction task. Conditioning inputs are segmentation maps that have to be embedded.

Figure 1: Architectural parts of Discrete Diffusion model adapted for segmentation. All architectures start with an input preprocessing block that handles (1) the embedding of segmentations, (2) the concatenation of the embedded noisy segmentation (a) with the conditioning tensor (b and c).

## 3.4 MOTIVATING EXAMPLE: THE RECTANGLE WORLD

In order to further motivate the need for properly modelling uncertainty in the segmentation task, we create an artificial image segmentation dataset. Each image contains two rectangles randomly filled with one of two colors, with equal probability. However, instead of mapping each color to a single category, it is mapped randomly and equiprobably to two categories. In each image, there are exactly four equally probable correct predictions since we use two rectangles per image, as shown in fig. 2a. We additionally add noise to the input image by flipping each pixel to a random color with $40\%$ probability.

We train a deterministic segmentation model and our generative diffusion model on the aforedescribed dataset for $100,000$ iterations. Both models, as expected, result into very crisp segmentations. However, as can be observed in fig. 2b the deterministic model completely misses half of the categories and only ever predicts two. In contrast, the generative model captures the uncertainty of the dataset and generates all possible classes even though the rectangles are still crisp and contain only a single category. Further details and qualitative results of this experiment are provided in appendix E.

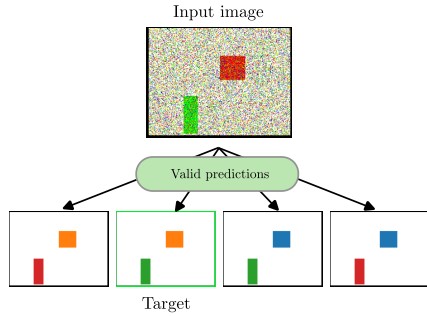

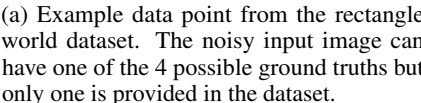

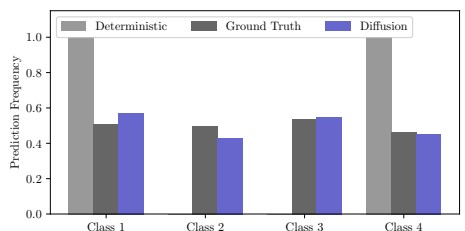

(a) Example data point from the rectangle world dataset. The noisy input image can have one of the 4 possible ground truths but only one is provided in the dataset.

(b) The predicted class frequency from 100 random inputs in the test set. We see that the deterministic model only generates 2 classes.

Figure 2: Fig. 2a shows an example from the rectangle world dataset. Fig. 2b showcases our results.

# 4 EXPERIMENTS

Building on the insights of the motivating example presented above, we conduct more extensive experiments using both simulated and real-world datasets.

For each task, we train a diffusion model and perform inference using only 10 diffusion steps, since we notice that performing more steps was not bringing significant quality improvements while noticeably impacting generation time. We also train a deterministic model in addition to the diffusion model. This model has the exact same architecture as the diffusion model, except it does not receive the 'current' noisy segmentation as input. This model naturally generates a single output, which corresponds to the maximum likelihood prediction for a given input.

The use of a deterministic model is generally ill-suited for ambiguous tasks, and even more for auto-regressive future prediction due to the possible error accumulation in the autoregressive process. However, despite these potential issues, we have found that generated sequences are generally of good quality, and we have therefore also reported 'deterministic' results for all experiments.

## 4.1 LUNG CANCER DATASET

### 4.1.1 DATASET

The first ambiguous task we aim to tackle is the segmentation of tumors from the medical image dataset LIDC (Armato III et al., 2011). This dataset contains volumetric lung CT scans, and we have adopted a pre-processing methodology identical to the one described in Kohl et al. (2018) so we can make meaningful comparisons to their results.

Four expert radiologists have individually annotated each scan for the presence of abnormalities. For a given scan image, the tumor annotations can vary significantly among the radiologists. By design, the training of conventional deterministic models cannot capture this variability. Segmentation diffusion models fit perfectly to this context, and allow sampling from the learnt distribution.

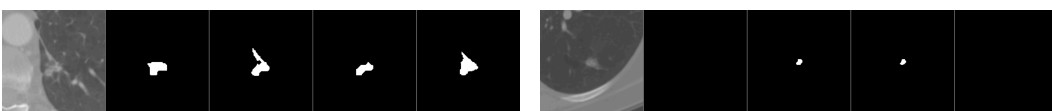

Figure 3: Two training examples from the LIDC dataset. The leftmost column are the lung scans, while the four next columns are the lesion masks as segmented by four different radiologists. These examples illustrate the substantial variability inherent in the dataset.

### 4.1.2 RESULTS

As mentioned previously, we train both a diffusion model and a deterministic model on this task. During training, we sample at random one of the four available ground truths to perform the opti-

mization step. To evaluate the performance of these models, we employ the Hungarian-matching based metric proposed by Kohl et al. (2019). This consists in sampling multiple segmentation masks from the model and using linear assignment to match each mask with the ground truth it has the highest IoU with. In this case, we sample 16 segmentations. Note that we assign the same number of segmentations to each ground truth annotation. For more details regarding the metric computation, see appendix B.2.

| Hungarian Matched IoU (%) | Full Test Set | Subset B |
| --- | --- | --- |
| H. P. Unet (Kohl et al., 2019) | 53 | 47 |
| Deterministic Model | 43.8 | 47.5 |
| Diffusion Model | 54 | 54.9 |

Table 1: Hungarian Matched IoU (%) for the Hierarchical Probabilistic Unet, a deterministic model, and our proposed diffusion model, all computed using 16 samples. The diffusion model is on-par on the full test set, and noticeably better when looking at subset B. Subset B comprises only the scans where all 4 radiologists found a lesion.

We report the results in table 1 for the 2 models, alongside the results of the Hierarchical Probabilistic Unet (HPU) as reported in Kohl et al. (2019). We also show example generations in fig. 4. Our results are on the full test set as well as on subset B, a subset comprising of all scans where all four experts agree that there is some lesion. Examining the results on subset B is interesting as they are more related to the model's proficiency in capturing shape variations. Indeed, an incorrect prediction, even if small, of a lesion where none exists results in a zero IoU score. This edge case has a strong influence on the overall HMIoU score. When looking at subset B, the presence of a lesion is generally clear, so most model generations should contain a non-empty segmentation and the metric is mostly influenced by the model's ability to accurately outline lesion shapes and variations.

With a 1% improvement, the diffusion model is marginally better than the original HPU paper on the full test set. However, on the subset B the diffusion model performs substantially better than HPU, which indicates our model's superior ability in capturing shape variations. Interestingly, on subset B the simple deterministic Unet model already performs as well as the HPU. This suggests that the HPU does not generate enough variations that properly cover the predictions of the physicians.

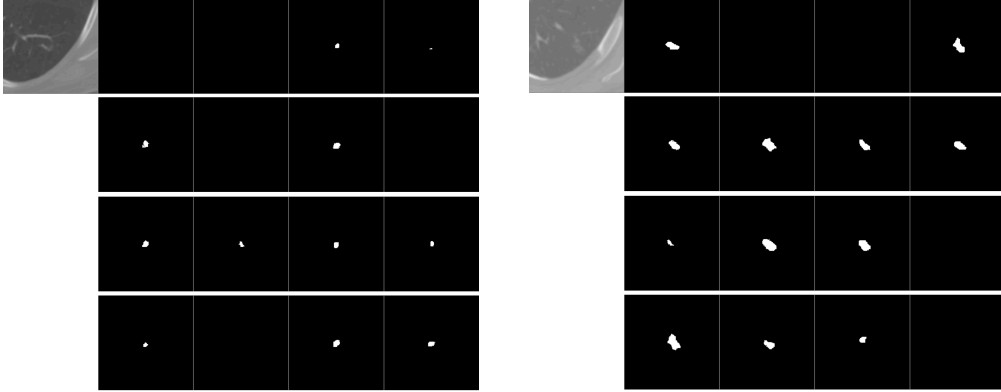

Figure 4: Qualitative results for two different lung scans on which we sampled the diffusion model 12 times. The top row has the input and the four ground truths. The three bottom rows show samples.

## 4.2 CAR INTERSECTION SIMULATOR

In order to clearly showcase the ability of these diffusion models to handle complex distributions with multiple modes, we design a dataset where each frame has a distinct and limited set of potential future scenarios. Precisely, this dataset is constructed such that the uncertainty between two successive frames can be represented as a Markov process. Because this transition is Markovian, we train the models only using one previous segmentation as conditioning input. Note that we perform

auto-regressive segmentation prediction, but that we evaluate the overall performance of the model using a trajectory prediction metric, which allows to better quantify how often we are able to sample the correct scenario.

To establish a strong baseline, we train a Transformer encoder model alongside deterministic and diffusion models. Since the segmentation space is too large to be handled by the transformer model directly (a sequence length of 65k), we use a VQVAE (Van Den Oord et al., 2017) to encode the segmentations into a latent space where the transformer generates its predictions. For more details about the architecture, please refer to appendix C.2.

### 4.2.1 SIMULATOR AND DATASET

The dataset is created using a custom bird's-eye-view car driving simulator that generates both an image and the corresponding segmentation map. In this simulator, the road layouts feature a single intersection, which can be either a roundabout or a three/four-way crossing. Each simulation features a random number of cars between 2 and 5. Each car belongs to one of four classes and moves along the roads with the same constant speed, kept identical across simulations. Additionally, its direction is clearly identifiable from the segmentation mask. Therefore, given the current segmentation, all potential routes a car can take are unambiguously predetermined. The intersection type determines the number of potential future routes a car may have, with a maximum of four different possibilities. For this reason, we use a single previous timestep as conditioning for all networks.

The final dataset consists of 4,992,000 training and 1000 validation examples generated by running the simulator for 20 steps and capturing the segmentation map of each frame. Thus, a training example consists of 20 successive segmentation maps of dimension 256x256.

### 4.2.2 RESULTS AND DISCUSSION

As this is a future prediction task, we use the auto-regressive scheme during inference to predict the segmentation of the next 20 frames. From the sampled sequence, we map out the overall trajectory of each car using a simple tracking algorithm and compute the Final Distance Error (FDE) between the last predicted position and the ground truth. For evaluation, we repeat this sampling scheme 10 times to get 10 distinct trajectories per car, from which we only select the trajectory exhibiting the lowest FDE. In table 2, we report the mean lowest FDE across 10 samples. We also compute the miss rate, defined as the fraction of trajectories whose FDE exceeds two, which implies the trajectory diverges significantly from the actual route. Please refer to appendices C.4 and C.5 for more details regarding our trajectory extraction procedure and the impact of the number of sample on the miss rate. The inference time per sample, recorded in milliseconds, is measured on an Nvidia V100 GPU.

| | FDE | Miss Rate | TIME (ms/spl) |
|---|---|---|---|
| Deterministic Model (36M) | 68 | 52.1 % | 70 |
| Transformer Encoder (50M) | 11.9 | 19.2 % | 3270 |
| Diffusion Model (36M) | 11.4 | 16.1 % | 580 |

Table 2: Results for trajectory predictions on the car simulator dataset, using the trajectory with lowest FDE among 10 samples. Both generative models solve the task, and the diffusion model performs best. The modest performance obtained by the deterministic model confirms the inherent ambiguity of the dataset. The high sampling time of the transformer encoder suggests that these models scale poorly to higher dimensionality. Lower is better in all columns.

Table 2 shows that the diffusion model is performing best, and the miss rate more specifically indicates that this model is most often predicting the correct trajectory compared to the transformer baseline. Additionally, contrary to the transformer model which samples each pixel sequentially and has to work in a learnt latent space, the diffusion model works in the image space and has an adjustable number of steps. As specified earlier, we only use ten steps in all experiments, which makes generation with diffusion about five times faster than with the transformer.

Interestingly, the deterministic model has a 52 % miss rate, which indicates that the one trajectory predicted by this model is correct about half the time. While confirming a degree of ambiguity in the

dataset, this miss rate shows that the deterministic model is still able to complete the task at least half of the time, and not degenerate the predicted segmentation as the auto-regressive process goes on.

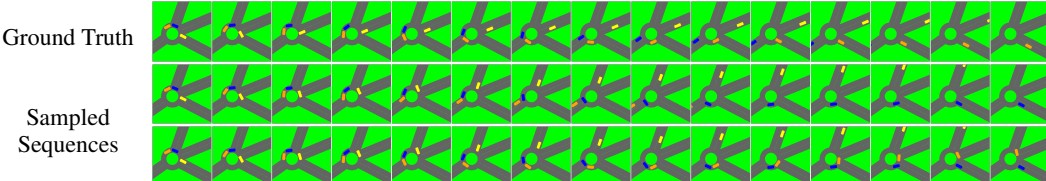

Figure 5: Qualitative results for sampling the diffusion model on the car intersection simulator. The top row is the ground truth, the two bottom rows are sampled sequences. We show only the fifteen first frames due to space constraints. The leftmost frame is the one used as first input to the model.

### 4.3 CITYSCAPES

Finally, we aim to tackle the task of segmenting first-person view road scene. The goal of this task is to predict the segmentation results of unobserved future frames. Short term future can generally be treated as a deterministic task, relying on the current state of the scene and motion models to estimate what will happen in the next few frames. In contrast, mid-term future is often a complex prediction problem. As we look further into the future, the task becomes more uncertain and the scene may diverge into different distinct scenarios depending on the behavior of other road users. For instance, a vehicle ahead may change lanes or a pedestrian may start crossing the road.

When we consider the potential combinations of all feasible behaviors for each agent present in the scene, it can yield dozens of distinct future outcomes. The ability to generate multiple samples of possible future segmentations allows to cover many of the different scenarios, and to address critical questions such as: '*Is there a scenario in which the child crosses the road ?*'. From this perspective, we believe it is reasonable to compare segmentation results obtained from existing methodologies with the best sample we can generate through our diffusion-based approach.

#### 4.3.1 DATASET

We conduct our experiments with the Cityscapes Dataset (Cordts et al., 2016), which was collected specifically for video-based segmentation research. It comprises a total of 5000 sequences, with 2975 sequences designated for model training, 500 for validation, and 1525 for testing. A sequence is made of 30 image frames, among which the 20th frame is manually annotated for semantic segmentation.

Following Lin et al. (2021), we perform short-term ($t + 3$) and mid-term ($t + 9$) prediction, and additionally report next-frame prediction ($t + 1$). To perform the short-term prediction, we try two configurations: we use the subsampled sequence $\{S_{t-6}, S_{t-3}, S_t\}$ as input and do one auto-regressive (AR) step similarly to Lin et al. (2021); we also show results using $\{S_{t-2}, S_{t-1}, S_t\}$ with 3 AR steps. To perform the mid-term prediction, we input the sequence $\{S_{t-6}, S_{t-3}, S_t\}$ and do 3 AR steps.

To extract the segmentation masks used as conditioning inputs, we run a pretrained segmentation network on previous timesteps. Specifically, we run the transformer-based Segmenter-B model from Strudel et al. (2021), pretrained on ImageNet and fine-tuned on Cityscapes.

#### 4.3.2 RESULTS AND DISCUSSION

In table 3, we employ a simple visual code to depict different prediction configurations. Each circle symbolizes one frame, and we arrange them linearly to represent the consecutive frames. The blue circles denote frames used as input, and the red ones the predicted frames. When performing multiple auto-regressive steps, we use one line for each step and stack them vertically.

For the diffusion model, we present results of using 1, 10 and 100 samples. When sampling multiple future segmentations, we calculate the mIoU based on the sampled segmentation that best matches the ground truth, as determined by the IoU metric. As motivated above, we argue that when the objective is to be able to account for all possible scenarios in order to capture the true one, it is fair to compare with the 'best' of multiple diffusion samples.

| mIoU (%) | | Short Term | | Mid Term |
|---|---|---|---|---|
| Saric et al. (2020) | N/A | 69.6 | N/A | 57.9 |
| Sović et al. (2022) | N/A | 70.2 | N/A | 58.5 |
| Lin et al. (2021) | N/A | 71.1 | N/A | **60.3** |
| Deterministic Model | **79.5** | 71.5 | 72.2 | 57.6 |
| Diffusion Model (1 spl) | 78.8 | 70.8 | 71.4 | 56.6 |
| Diffusion Model (10 spl) | 79.1 | 71.3 | 72.4 | 58.0 |
| Diffusion Model (100 spl) | 79.3 | **71.9** | **73.1** | 59.1 |

Table 3: Results for Cityscapes future segmentation tasks. In the header, each circle represents a time-step, blue ones are used as input and red ones are predicted. The diffusion model with multiple samples is competitive with current state-of-the-art networks. The deterministic model most of the times falls between 1-sample and 100-samples evaluation, confirming the ambiguity of the task. Results are in terms of mIoU percentage, where higher is better.

While the current best baseline (Lin et al., 2021) performance is strong, we outperform it in the short-term prediction when using multiple samples, and we come close in the mid-term prediction. Note that this baseline model relies on predicting intermediate image features, while we only use previous segmentation maps. We conjecture that our lower results in the mid term context compared to the deterministic model from Lin et al. (2021) is relatively imputable to the information loss of using segmentation maps as inputs. As their feature prediction strategy is orthogonal to our approach, we believe we could benefit from it and achieve more competitive results.

Additionally, it is interesting to note that the results of the deterministic model are almost always higher than 1-sample diffusion but lower than 100-samples diffusion. This observation aligns with our expectations, given that the deterministic model is specifically trained to output the most likely outcome, while the diffusion model stochastically samples from possible outcomes. Consequently, it is reasonable to anticipate slightly lower performance on average for the 1-sample diffusion results, as they only represent a single stochastic sample. However, if the task indeed contains ambiguity and the diffusion model can effectively sample from the different modes of the distribution, we would expect improved results on average when considering the best of 100 samples, as it is likely that at least one of them will more closely match to the ground truth. And we indeed see that as we predict further in the future and uncertainty increases, the performance gap between the deterministic model and the 100-samples diffusion model also widens.

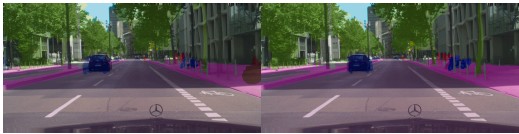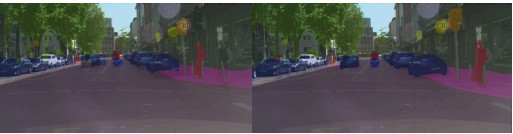

Figure 6: Qualitative results for two Cityscapes mid-term future segmentation scenarios at $T + 9$. The left column shows what would happen if we were to simply copy the last segmentation map, the right column shows one sample from our diffusion model.

## 5 CONCLUSION

This paper has explored the application of discrete diffusion models in addressing the challenges posed by non-deterministic and high-uncertainty situations in different segmentation tasks, specifically in the domains of future prediction and medical image segmentation. In the context of future segmentation prediction, we have presented an innovative approach that leverages discrete diffusion models using only previous segmentations as input within an auto-regressive framework. Our experiments show that these discrete models offer a promising avenue for performing segmentation in such ambiguous contexts. A relevant future direction for our research is to find ways to force diversity in the sampling process so that we can reduce the need for repetitive sampling to cover the whole space of the distribution.

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

## A  MODEL ARCHITECTURE DETAILS

For all tasks, the network architecture is a U-Net from Wang (2023), both for the deterministic and diffusion model, and the only difference is the input processing block. The segmentations are embedded in a 32 dimensional space learnt during training in all experiments. We use 5 contractions (down) and 5 expansions (up) layers, the initial dimension is 32 and the dimension multipliers are $(1, 2, 2, 4, 8, 16)$.

## B  LIDC EXPERIMENT DETAILS

### B.1  TRAINING DETAILS

As mentioned in section 4.1.1, the pre-processing follows Kohl et al. (2018). In particular, it involves the transformation of the original 3D scans into 2D image crops. These are then re-sampled to 180x180 pixels and centered around the scan abnormality location. We omit the inclusion of small abnormalities measuring less than 3mm in this process, as they are often considered clinically insignificant. This results in 8843 images for training, 1993 for validation and 1980 for testing. In addition to the above, the data preparation pipeline for training consists in: randomly selecting one of the 4 ground-truths, resizing image and label to 128x128 pixels using nearest-neighbor interpolation, randomly applying horizontal flip, and normalizing the images with a mean of 0.41 and a standard deviation of 0.21.

We train both models with batch-size 64 for 10,000 epochs, using a prediction loss weight $\lambda = 1e^{-3}$ for the diffusion model. We use AdamW with a learning rate of $1e^{-3}$ and weight decay of $1e^{-4}$, and a polynomial learning rate scheduler.

### B.2  METRIC

As described in section 4.1.2, we evaluate the performance using the Hungarian-matching based metric proposed by Kohl et al. (2019).

The idea behind this metric is to measure the alignment between the model's predictions' distribution and the experts' predictions' distribution. To do so, we determine the best matching between the sampled segmentations and the ground truth annotations using linear assignment on the IoU scores.

In practice, we first sample a number of segmentation masks that is a multiple of the number of ground truths. In the case of LIDC, there are 4 ground truths and we sample 16 segmentations. Note that sampling more segmentations does not increase the score but reduces its variance. Then, we compute the IoU between every pair of samples and labels. We use these scores to perform linear assignment, assigning 4 samples to each of the 4 ground truth labels. Finally, we average the IoU of the all the pairs that have been assigned together. Figure 7 visually explains this process when using 4 samples.

### B.3  QUALITATIVE RESULTS

On fig. 8, we show more qualitative results of LIDC segmentation samples. Samples have been vertically aligned under the ground truth they were matched with by the HMIoU computation. The examples on the left are showing failure cases where the shapes are only partly correct and their frequency is not matching the labels' frequency. The examples on the right shows success cases where complex shapes variation are properly captured.

## C  CAR SIMULATOR EXPERIMENT DETAILS

### C.1  TRAINING DETAILS

We train all models for 1 epoch and there are no data augmentation.

The diffusion model use batch-size 25, using a prediction loss weight $\lambda = 1e^{-3}$. We use AdamW with a learning rate of $1e^{-3}$ and weight decay of $1e^{-4}$, and a polynomial learning rate scheduler.

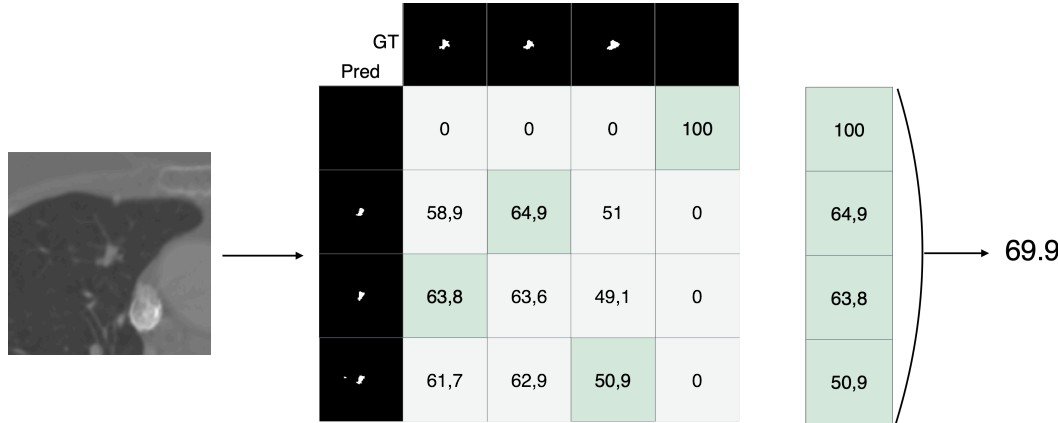

Figure 7: Visual explanation of HMIoU. We sample 4 segmentation masks (Pred), compute their IoU with each of the ground truth (GT) labels, find the best linear assignment, and average the IoU scores of the assigned pairs.

The deterministic model uses batch-size 40. We use AdamW with a learning rate of $3e^{-4}$ and weight decay of $1e^{-4}$, and a polynomial learning rate scheduler.

The Transformer-Encoder model uses batch-size 40. We use AdamW with a learning rate of $1e^{-3}$ and a polynomial learning rate scheduler.

## C.2 TRANSFORMER-ENCODER ARCHITECTURE

As generative baseline for this new task, we use a transformer encoder architecture with 16 encoder layers, and feature dimension 512. Since the $256 \times 256 = 65536$ pixel space is too large to process, we use a VQ-VAE to embed the input segmentations into a latent space using 256 codes. Given the 256 latent codes from the input segmentation, the transformer is sequentially generating the 256 codes of the next frame's segmentation.

**VQVAE Architecture** The VQVAE consists of an Encoder, a Decoder and a Quantizer.

The VQVAE encoder consists of:

- An embedding layer mapping the segmentation classes to a 16-channels continuous space.

- A convolutionnal stem (in_channels=16, out_channels=128, kernel_size=2, stride=2)

- A sequence of 5 ConvNext blocks with 128 channels, interlaced with 3 averaging pooling layers (kernel_size=2).

- A linear projection layer (in_channels=128, out_channels=32)

The VQVAE quantizer is an EMA Vector Quantizer with a dictionary of 512 vector codes of dimension 32. The EMA quantizer uses exponential moving averages to update the embedding vectors instead of using an auxiliary loss.

The VQVAE decoder consists of:

- A linear projection layer (in_channels=32, out_channels=128)

- A sequence of 5 ConvNext blocks with 128 channels, interlaced with 3 bilinear upsampling layers (scale_factor=2).

- An output layer mapping the output back to the 16-channels input embedding space.

This VQVAE is trained over $100,000$ samples using a batch-size of 10. We use AdamW with a learning rate of $1e^{-3}$ and weight decay of $1e^{-4}$, and a polynomial learning rate scheduler.

## C.3 QUALITATIVE RESULTS

On fig. 9b, we show more qualitative results for intersections with multiple possible scenarios.

## C.4 IMPACT OF THE NUMBER OF SAMPLES ON THE MISS RATE

The Figure 10 shows the evolution of the miss rate, which measures how frequently the correct trajectory is absent among the sampled trajectories, with respect to the number of samples used to compute the miss rate. Each sample allows to get one new possible trajectory per car in the scenario, extracted from the segmentation sequence using our trajectory extraction procedure. Some of these trajectories may be very similar to each other, but fig. 10 shows that the more we sample, the more likely we are to generate the correct trajectory. The notable reduction in the miss rate with each additional sample indicates that the models are indeed able to cover a substantial portion of the trajectory space.

## C.5 TRAJECTORY EXTRACTION AND COMPARISON

To extract car trajectories from a sequence of segmentation maps, we implement the following procedure:

For all frames:

- Identify all connected components and discard those that are too small,
- Determine the center of mass for these components, which will represent the car locations.

For each pair of consecutive frames:

- Calculate the squared distance matrix between car locations in the first frame and those in the subsequent frame,
- Compute the optimal pairing of car locations from the first frame with those from the second, utilizing linear sum assignment,
- Remove from the pairing all pairs of cars that are too far from each other,
- For car locations that remained not paired, consider them as entering or exiting the scenario.

By keeping track of all paired car locations while processing all consecutive segmentations, we can derive the complete trajectory.

In our car intersection simulator experiments, we enhance robustness by excluding trajectories of cars that don't begin at the start of the simulation, as all simulated cars are present in the segmentation right from the beginning of the simulation.

To compare trajectory extracted from the ground truth sequence and a sampled sequence, we use the fact both sequences share the same initial frame. Hence, we simply align trajectories based on the cars' initial positions.

## D CITYSCAPES EXPERIMENT DETAILS

## D.1 TRAINING DETAILS

On the Cityscapes experiments, we use 3 previous segmentations all embedded in 32 dimensions, the conditioning tensor is therefore 96 dimensional. The 3 previous segmentation have a constant time spacing between each of them. However, during training, we have varied this time spacing for each training sample.

More specifically, during training we randomly sample a spacing $\tau \in \{1, 2, 3\}$ and we use the segmentations $(S_{t-3\tau}, S_{t-2\tau}, S_{t-\tau})$ as input to the models. This allows us to use such a trained model with any small constant spacing $\tau$ in inference, and in fact all the results reported for the deterministic and diffusion model in tables 3 and 4 are computed using a single model.

We train the diffusion model for 1000 epochs with a batch-size 3 and a prediction loss weight $\lambda = 0.1$. We use AdamW with a learning rate of $3e^{-4}$, weight decay of $1e^{-4}$ and a polynomial learning rate scheduler.

We train the deterministic model for 1000 epochs with a batch-size 1. We use AdamW with a learning rate of $1e^{-4}$, weight decay of $1e^{-4}$ and a polynomial learning rate scheduler.

### D.2 MOVING OBJECT RESULTS

In table 4, we report results for Cityscapes computed on Moving Objects only. The patterns we observe are similar to those from table 3, with the deterministic model performances being always between the 1-sample and 100-samples diffusion model results.

The diffusion model using multi-samples suffers from a much smaller performance drop compared to all other models, including Lin et al. (2021). This highlights the ability of the model to capture different scenarios.

| mIoU (%) | | | | |
| --- | --- | --- | --- | --- |
| | | Short Term | | Mid Term |
| Saric et al. (2020) | N/A | 67.7 | N/A | 54.6 |
| Sović et al. (2022) | N/A | 69.0 | N/A | 55.9 |
| Lin et al. (2021) | N/A | 69.2 | N/A | **56.7** |
| Deterministic Model | 79.7 | 70.5 | 71.3 | 53.6 |
| Diffusion Model (1 spl) | 79.0 | 69.9 | 70.0 | 52.5 |
| Diffusion Model (10 spl) | 79.5 | 70.9 | 71.9 | 54.5 |
| Diffusion Model (100 spl) | **80.0** | **71.5** | **72.7** | 56.0 |

Table 4: Results for Cityscapes future segmentation tasks on **Moving Objects**. In the header, each circle represents a time-step, blue ones are used as input and red ones are predicted. The diffusion model with multiple samples is competitive with the current state-of-the-art network. The deterministic model most of the times falls between 1-sample and 100-samples evaluation, confirming the ambiguity of the task. Results are in terms of mIoU percentage, where higher is better.

### D.3 QUALITATIVE RESULTS

We show more samples of our diffusion model for future segmentation on Cityscapes in fig. 11.

## E MOTIVATING EXAMPLE DETAILS

The dataset consists of images of size $240 \times 320$ pixels, each containing two rectangles filled in either green or red. Since it is an artificial dataset, we have a random seed for training and a different random seed for evaluation. To make the problem slightly more challenging, we randomly swap 40% of the pixels with random colors as can be seen clearly in Fig 12.

We train both models for $100, 000$ iterations with a batch size of 16. The Unet has 2.8M parameters and is the same for both the diffusion model and the deterministic one, with the exception of the first layer that embeds the segmentation mask or the image respectively.

In addition to the results shown in section 3.4, we also measure the ability of our generative models to create all possible classes given a single input image. In fig. 13 we show the frequency with which each class was predicted for each rectangle when generating 100 images from a single input. We observe that all classes are predicted with probability close to 50%, same as they appear in the dataset.

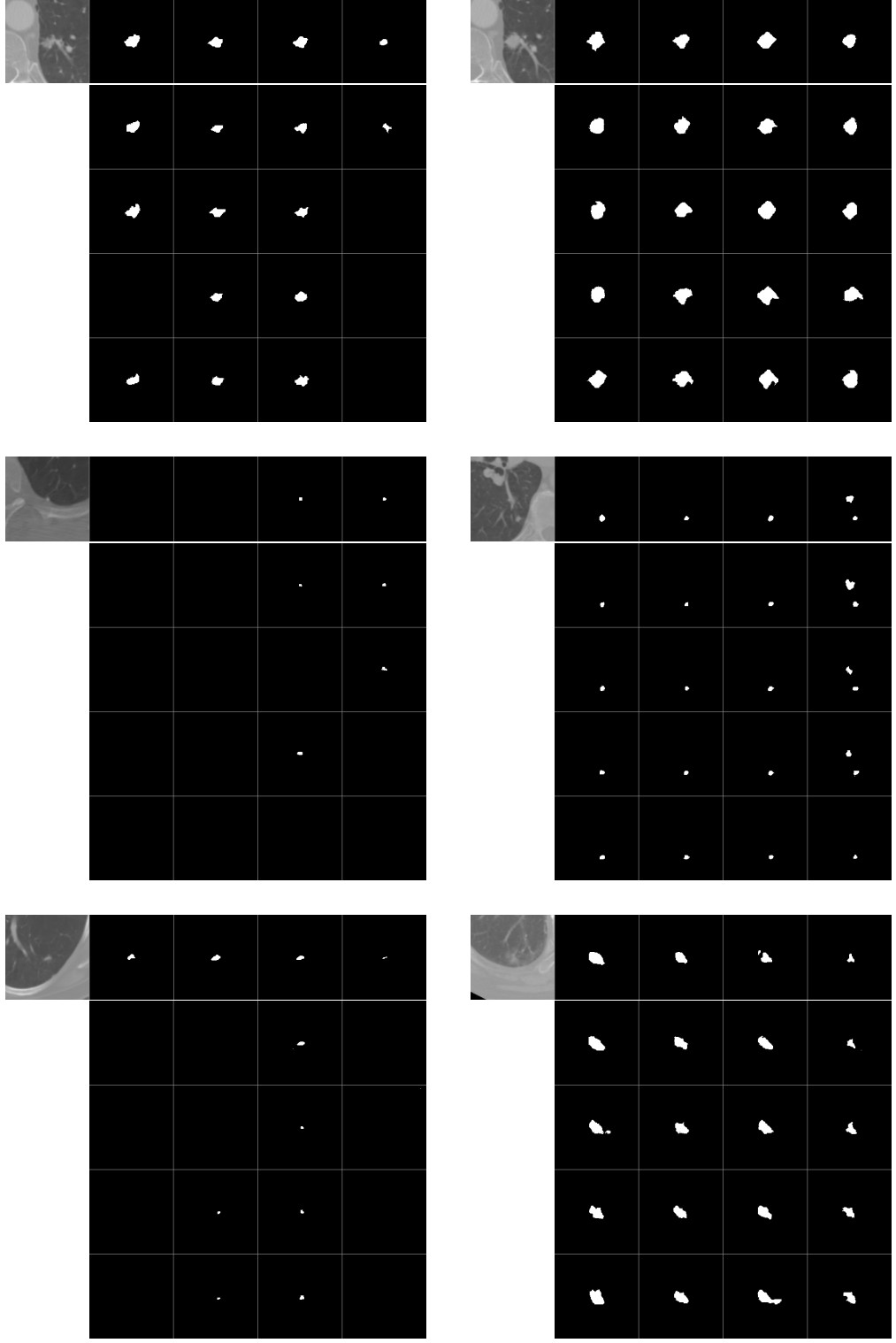

Figure 8: Qualitative results for 6 different lung scans on which we sampled the diffusion model 16 times. For each example, the top row has the input scan and the four ground truths. The four bottom rows show samples. The examples on the left are failure cases, the examples on the right are well capturing shape variations.

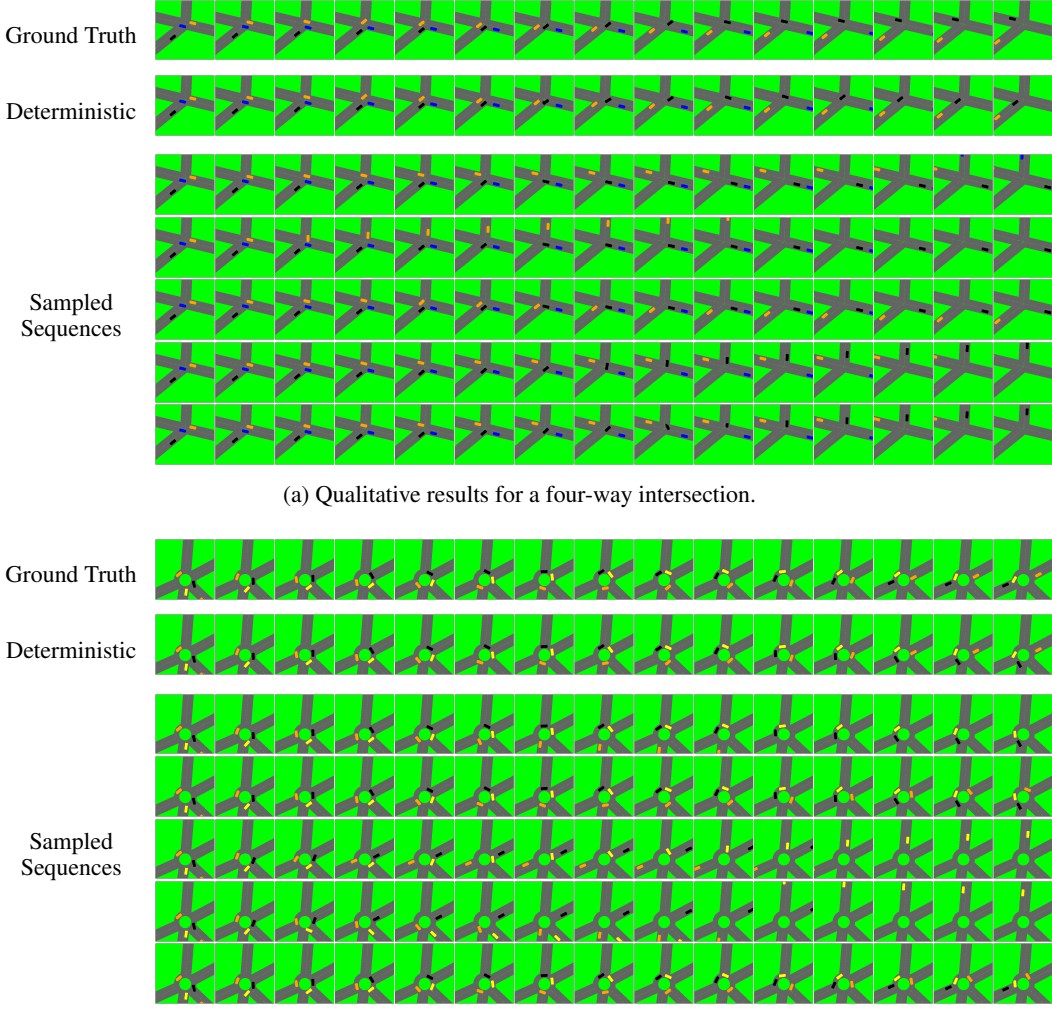

(a) Qualitative results for a four-way intersection.

(b) Qualitative results for a four-way roundabout.

Figure 9: Qualitative results on the car intersection simulator. From top: The first row is the ground truth, the second row is the result from the deterministic model, the five next rows are sampled sequences. We show only the fifteen first frames due to space constraints. The leftmost frame is the one used as first input to the model.

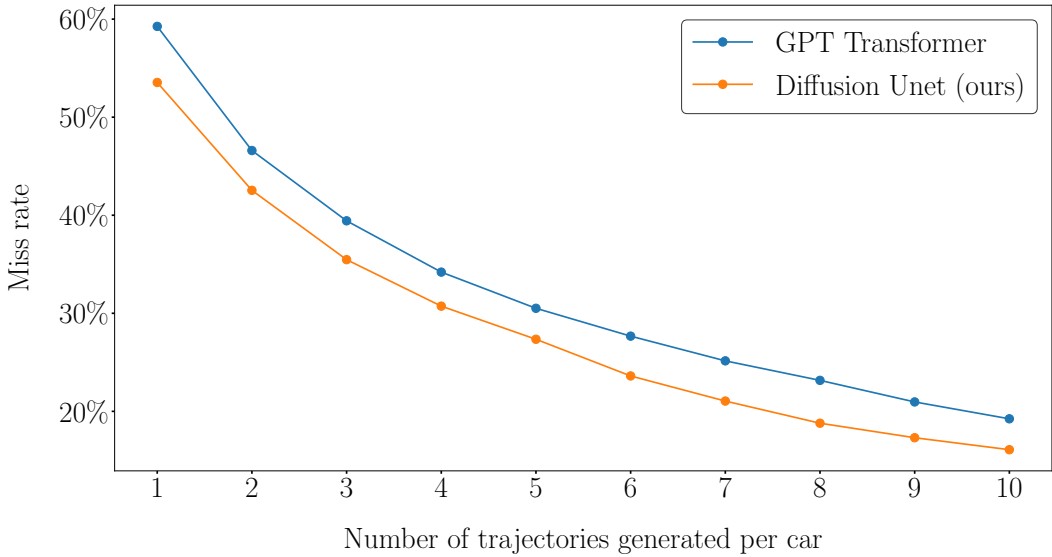

Figure 10: Evolution of the miss rate with the number of samples used during evaluation, both for the diffusion model (orange) and the Transformer-Encoder model (blue). Performing multiple samplings clearly allows to cover more scenarios and generate the real trajectory more frequently.

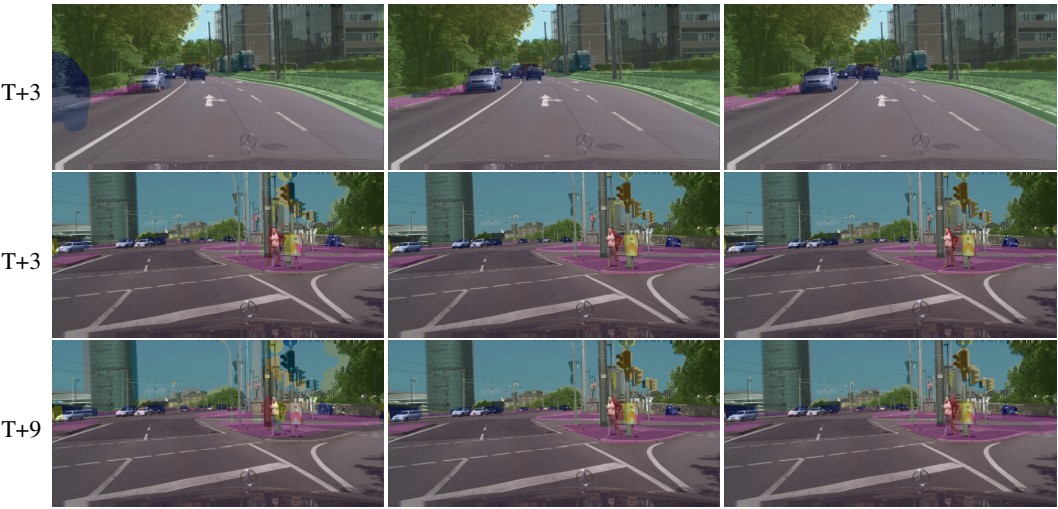

Figure 11: Qualitative results for Cityscapes short-term ($T + 3$) and mid-term ($T + 9$) future segmentation scenarios. The left column shows what would happen if we were to simply copy the last segmentation map, the next columns on the right shows samples from our diffusion model.

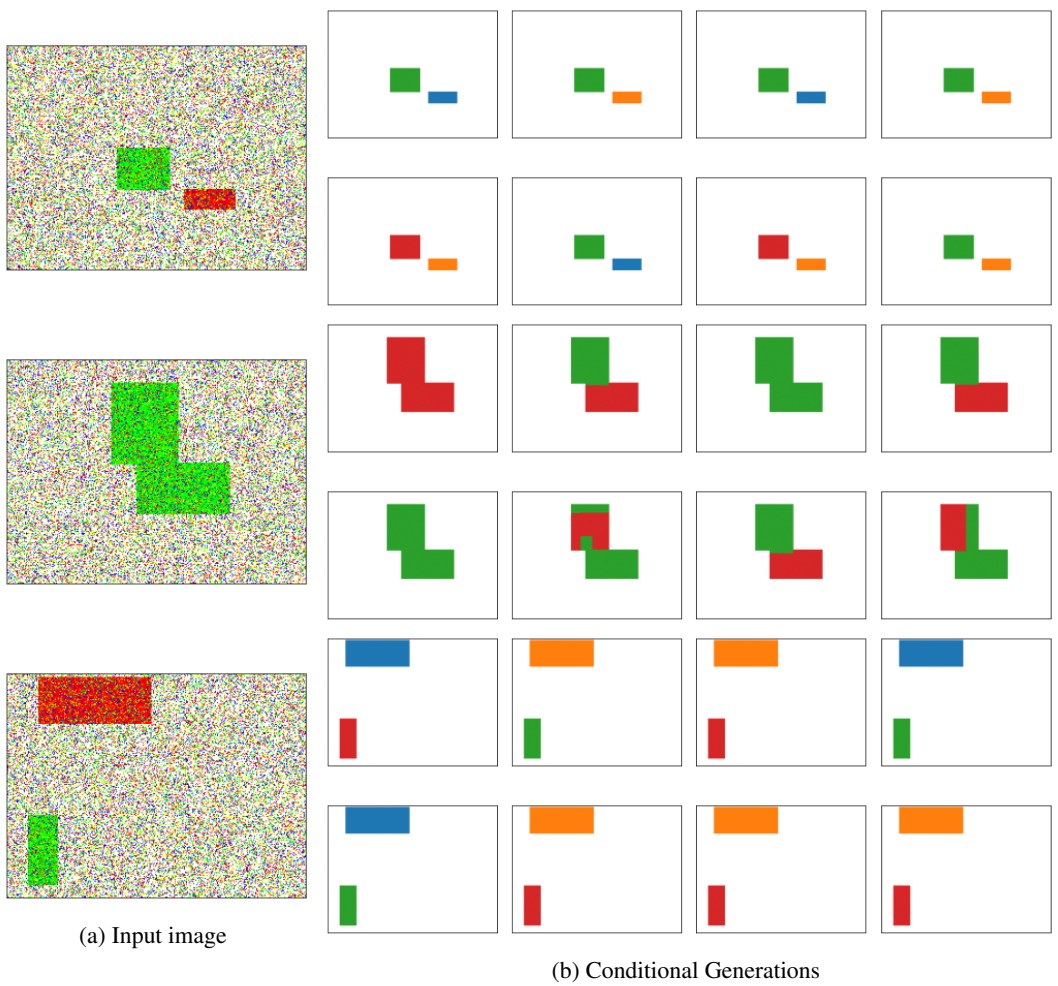

(a) Input image

(b) Conditional Generations

Figure 12: Example input images and 8 generations from the diffusion model. We observe that all classes have been generated for the input rectangles.

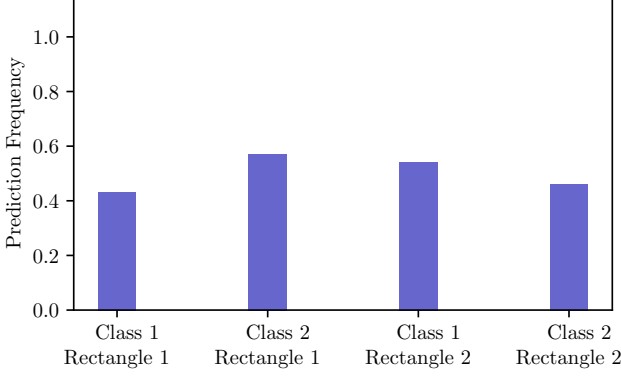

Figure 13: We show the predicted class frequency from 100 generated samples from our diffusion model conditioned on the same input image. All categories are generated with equal probability.

