# OpenReview forum: "Segmenting the Unknown: Discrete Diffusion Models for Non-Deterministic Segmentation"
_ICLR.cc/2024/Conference — Submitted to ICLR 2024_

### Official Review · Reviewer_ofDT · 2023-10-29

**Soundness:** 2 fair
**Presentation:** 2 fair
**Contribution:** 2 fair
**Rating:** 3
**Confidence:** 5

**Summary:**

The paper introduces discrete diffusion models to address ambiguity and uncertainty in semantic segmentation, specifically for applications in oncology and autonomous driving. The authors propose a unified solution for two tasks: future prediction and medical image segmentation. This solution leverages the discrete diffusion framework to model segmentation annd prediction uncertainty. They also introduce an auto-regressive diffusion framework for future forecasting. Experimental evaluations were conducted on a Lung Cancer medical Imaging Dataset (LIDC) and two future prediction tasks, demonstrating the efficacy of their proposed generative framework.

**Strengths:**

- The paper presents a unified approach to handle future prediction and image segmentation, reducing the need for distinct solutions for each task.

- Experimental evaluations show that their model consistently outperforms equivalent deterministic models in all tasks. Additionally, the proposed generative framework surpasses existing VAE-based methods on LIDC and is on par with state-of-the-art methods on Cityscapes future prediction.

**Weaknesses:**

- There is a heavy reliance on existing ideas, with incremental adaptation to an existing discrete diffusion model. The paper simply employs input conditioning via concatenation to adapt the generative model for segmentation. Moreover, autoregression for future prediction is just conditioned on past segmentations, leaving out crucial technical details that would illuminate the depth of the contribution.

- The paper did not explicitly define the type of uncertainty being captured, leaving ambiguity between aleatoric (data) and epistemic (model) uncertainty.

- While the paper compares its approach to deterministic methods, it lacks comprehensive comparisons with existing work in uncertainty quantification in semantic segmentation.

- Results show samples of the posterior, yet uncertainty is not explicitly quantified.

**Questions:**

- Given the lack of clarity on the type of uncertainty captured, can the authors specify whether it is aleatoric or epistemic uncertainty?

- The title suggests the capability to segment unknown classes. Can the authors clarify this claim?

- The paper discusses the potential for forcing diversity in the sampling process. Can the authors elaborate on possible methods to achieve this?

- Why is there a lack of comprehensive comparisons with existing uncertainty quantification methods in semantic segmentation?

- With the heavy reliance on existing ideas, can the authors provide further technical details or novel contributions that differentiate their approach from previous work?

- Considering the importance of understanding and reporting both types of uncertainties, why was this aspect not heavily emphasized in the paper? Furthermore, can the authors shed light on how uncertainty is being evaluated and calibrated?

---

> ### Author Response · Authors · 2023-11-21
>
> We thank you for your comments.
> We wanted to emphasize that, while one can see some connections between our work and uncertainty estimation, our work is not about uncertainty estimation. We introduce a concrete novel way to leverage discrete generative diffusion models for segmenting ambiguous contexts, and we assess their performance. We expand more on that point below.
>
> ---
> ## Detailed Answers:
>
> > Given the lack of clarity on the type of uncertainty captured, can the authors specify whether it is aleatoric or epistemic uncertainty?
>
> We consider aleatoric uncertainty, as we use experimental setups where the input has multiple possible valid segmentations.
>
> > The title suggests the capability to segment unknown classes. Can the authors clarify this claim?
>
> No, we do not make such a claim. The ”unkwown” refers to the ambiguity present in the contexts we consider. In
> particular, when doing future prediction, the future state is unknown, yet
> we propose a way to segment it.
>
> > The paper discusses the potential for forcing diversity in the sampling
> process. Can the authors elaborate on possible methods to achieve this?
>
> In our concluding remarks, we indeed identified encouraging diversity as a
> future research direction. A feasible initial approach would be to slightly
> penalize the previously predicted classes when generating the new segmentation. This serves as a baseline in our ongoing research.
>
> > Why is there a lack of comprehensive comparisons with existing uncertainty quantification methods in semantic segmentation?
>
> > Considering the importance of understanding and reporting both types of
> uncertainties, why was this aspect not heavily emphasized in the paper?
> Furthermore, can the authors shed light on how uncertainty is being evaluated and calibrated?
>
> We will add a section in our literature review on the most related uncertainty estimation methods for segmentation.
> **However, uncertainty estimation is not the focus of our work**. We do not evaluate it.
> We present a concrete way to leverage discrete generative diffusion models for ambiguous contexts, and we assess whether these models follow the data distribution and have the capacity to generate different predictions.
> We have chosen our experimental setup to be contexts where there is true uncertainty so that multiple different predictions could be equally valid. Estimating the model uncertainty in these contexts would be a proxy for estimating the dataset uncertainty, but we believe it would not reveal more useful information about the model.
>
> > With the heavy reliance on existing ideas, can the authors provide further technical details or novel contributions that differentiate their approach from previous work?
>
> Our work builds on existing ideas, but we propose an adaptation of these to segmentation, both as an image-to-segmentation model with LIDC and as an autoregressive segmentation-to-segmentation model with our two other experiments.
> We proposed these modifications and implemented and evaluated the use of discrete diffusion segmentation models in 3 diverse ambiguous setups to showcase their viability for this purpose.
> In fact, a very recent work [1] proposes a quite similar method, but applied only on LIDC, and was recently accepted at ICCV.
>
> We also refer you to our answers to Reviewer 1 (qH1j) regarding discussions with some similar and related works.
>
> [1] Stochastic Segmentation with Conditional Categorical Diffusion Models (https://arxiv.org/abs/2303.08888)

---

> > ### Comment · Reviewer_ofDT · 2023-11-22
> >
> > Thank to the authors for their responses to the concerns I initially raised. In light of these clarifications, and also considering the perspectives presented in other reviews, I have decided to revise my evaluation and raise my score for your submission.

---

### Official Review · Reviewer_27GV · 2023-11-01

**Soundness:** 3 good
**Presentation:** 2 fair
**Contribution:** 2 fair
**Rating:** 5
**Confidence:** 4

**Summary:**

This paper introduces discrete diffusion models to capture uncertainty in semantic segmentation, with application in both oncology and autonomous driving. Unlike prior approaches that tackle these tasks in distinct ways, the proposed method formulates both as estimating a complex posterior distribution over images, and presents a unified solution that leverages the discrete diffusion framework. The contributions include the adaptation of discrete diffusion for semantic segmentation to model uncertainty and the introduction of an auto-regressive diffusion framework for future forecasting. Experiments have been conducted on both medical imaging data and real-world future prediction tasks to demonstrate the superiority of the proposed generative framework over deterministic models and its competitive performance compared to methods specific to these domains separately.

**Strengths:**

- The idea of presenting a unified solution for both future prediction and medical image segmentation is interesting, by leveraging diffusion models.
- This paper proposes the first method to model the uncertainty of predictions using discrete diffusion models in semantic segmentation.
- The proposed method is quite straightforward to follow.

**Weaknesses:**

- The experimental section is limited. I expect to see the results of more baselines. For example, how the proposed method is compared with GAN-based methods?
- There are lots of works regarding uncertainty estimation in (semantic) segmentation [1,2,3], just list a few. I would like to see the authors do a thorough review of these kinds of methods and provide a comprehensive comparison with the proposed method.
- Beyond the segmentation evaluation metrics, I am expecting the see more empirical results regarding uncertainty estimation. Some of the metrics can be found in [4,5].

[1] Nair, Tanya, et al. "Exploring uncertainty measures in deep networks for multiple sclerosis lesion detection and segmentation." Medical image analysis 59 (2020): 101557.

[2] Fleuret, Francois. "Uncertainty reduction for model adaptation in semantic segmentation." Proceedings of the IEEE/CVF Conference on Computer Vision and Pattern Recognition. 2021.

[3] Czolbe, Steffen, et al. "Is segmentation uncertainty useful?." Information Processing in Medical Imaging: 27th International Conference, IPMI 2021, Virtual Event, June 28–June 30, 2021, Proceedings 27. Springer International Publishing, 2021.

[4] Moon, Jooyoung, et al. "Confidence-aware learning for deep neural networks." international conference on machine learning. PMLR, 2020.

[5] Li, Chen, Xiaoling Hu, and Chao Chen. "Confidence Estimation Using Unlabeled Data." International Conference on Learning Representations. 2023.

**Questions:**

N/A

---

> ### Author Response · Authors · 2023-11-21
>
> We thank you for your comments, and for pointing to uncertainty estimation literature.
> We are pleased that you noted the simplicity of our solution and its ability to work in diverse settings, and that you found it interesting.
> We wanted to emphasize that, while one can see some connections between our work and uncertainty estimation, our work is not about uncertainty estimation. We introduce a concrete novel way to leverage discrete generative diffusion models for ambiguous segmentation contexts, and we assess their performance. We expand more on that point below.
>
> ---
> ## Detailed Answers:
>
> > The experimental section is limited. I expect to see the results of more
> baselines. For example, how the proposed method is compared with GAN-based methods?
>
> We will add more related works on LIDC experiments.
> To our knowledge, no GAN-based method has been
> applied to this problem, and our work focuses on showing the ability of
> diffusion models to handle ambiguity. GANs are known to have mode
> collapse issues when learning distribution and can additionally be particularly hard to train.
> While it would be interesting to check how a particular GAN architecture would perform in our experimental setup, this is
> not the focus of this study and could be viewed as another contribution in itself.
>
> > There are lots of works regarding uncertainty estimation in (semantic)
> segmentation [1,2,3], just list a few. I would like to see the authors do
> a thorough review of these kinds of methods and provide a comprehensive comparison with the proposed method.
>
> Thank you for the pointers to relevant literature! We will definitely add a section to our literature review of the most related uncertainty estimation methods for segmentation. We want to emphasize though that the focus of our work is not on uncertainty estimation, but rather on presenting a concrete way to leverage discrete generative diffusion models for ambiguous contexts, and on assessing whether these models are able to follow the data distribution.
>
> > Beyond the segmentation evaluation metrics, I am expecting the see more
> empirical results regarding uncertainty estimation. Some metrics
> can be found in [4,5].
>
> As said above, we focus on the segmentation task and the capacity of the network to generate different predictions.
> We have chosen our experimental setup to be contexts where there is true uncertainty, so that multiple different predictions could be equally valid. Estimating the model uncertainty in these contexts would be a proxy for estimating the dataset uncertainty, but we believe it would not reveal more useful information about the model. However, any metric measuring the model uncertainty at the image level to assess the model confidence on a specific image could still be applied. In the case of the LIDC dataset, we could consider adding a correlation analysis of the model uncertainty per image with the ground-truth variations.

---

> ### Comment · Reviewer_27GV · 2023-11-23
>
> If the authors argue that this submission is about using the diffusion model for segmentation instead of uncertainty estimation, I would suggest the authors reorganize the paper and clarify the point. And also, if this is the case, the value of this submission to me is even more limited.

---

### Official Review · Reviewer_qH1j · 2023-11-01

**Soundness:** 3 good
**Presentation:** 3 good
**Contribution:** 2 fair
**Rating:** 6
**Confidence:** 3

**Summary:**

The paper presents an approach to apply discrete diffusion models to model uncertainty for both semantic segmentation and future forecasting semantic segmentation. The authors evaluate their method on both simulated data an real data and claim competitive performance.

**Strengths:**

The work mostly clearly contextualizes and motivates its approach in relation to prior work. The work cleverly transfers existing methods to new problems. \
The method is evaluated for multiple settings, showing that it is versatile and not restricted to a single problem setting, and for multiple datasets. The selection is motivated well. \
The supplement provides thorough information regarding experimental details.

**Weaknesses:**

A number of results with better performance for the LIDC dataset are missing, giving the impression that the results are sota (see Table 1 in [1]) \
A prior work [1] has previously proposed the use of discrete diffusion for handling uncertainty in semantic segmentation. This somewhat limits the novelty, however I think the two works can still be counted as concurrent work. \
A more detailed qualitative evaluation for Cityscapes would have been interesting. Where does the method perform well, where does it show weaknesses, does it learn something about the movement of other entities in the scene, where does the performance improvement from 1 to 100 samples evaluation come from? \
The work shows in the car simulator dataset that the method does generate a variety of future scenarios, however it is not shown for the more complex Cityscapes dataset. Showing it quantitatively is of course difficult with the existing data, however it is also not clearly shown qualitatively. \
For scenarios like the mentioned "Is there a scenario in which the child crosses the road?" to be applicable real-time performance is essential. The inference time for the method on the Cityscapes data is not mentioned.

[1] https://arxiv.org/abs/2303.08888

**Questions:**

Questions

Regarding the motivating example, how is it defined which rectangle is "rectangle 1" and which is "rectangle 2"? In other words, given an image, how can I distinguish the two categories of "the rectangles have different classes"? Secondly, what are the two categories the deterministic model predicts? \
[1] was made public earlier this year. Can you please elaborate on the differences to the work at hand? \
Chen et al. (2022a) (referenced by you) was made public last year. While they do not show results regarding handling ambiguity, at first glance it seems to be applicable. Does the approach at hand have specific properties that would make it more advantageous for this task compared to their approach? \
For clarification, in 4.2.2: For each car in each validation example 10 samples are generated. The best of the 10 samples is selected according to FDE. 84% of those FDE values are less then 2 (a "hit"). And then the mean of the best FDE values (hit and miss) is computed. Is this correct? I am not sure the mean is very informative then as the distribution seems to be quite skewed.

General Notes

There seems to be quite a bit of interesting information in the supplement that is never referenced in the main paper, e.g., the MO results for Cityscapes, making it difficult for the reader to be aware of it. \
In Sec 3.1/2 it is not always fully clear to me what parts are from Austin et al. and what parts are adaptations by the authors.

Most issues have been addressed satisfactorily in the Discussion Phase, thus I've updated my rating from 5 to 6.

---

> ### Author Response · Authors · 2023-11-21
>
> We thank you much for your comments and pointing to relevant literature we missed!
> We are happy that you noted our diverse experimental setting and found it was well-motivated.
> We took into account your general notes: we will better reference the
> interesting supplementary results and clarify the difference with Austin et al.
> Additionnaly, we will include in the paper our discussion below regarding the Cityscapes results.
>
> ---
> ## Detailed Answers:
>
> > Regarding the motivating example, the definition of the rectangles and the prediction of the deterministic model:
>
> *Differentiation of Rectangles*: The rectangles within the images
> are not assigned fixed identifiers such as ”Rectangle 1” or ”Rectangle 2”.
> For each image, we create 2 rectangles, randomly assigned to one of 4
> classes. However, the classes 1 and 2 have the same color in the input
> image, just as the classes 3 and 4. In practice, on the generation plots,
> the color mapping we use for each class is: {1: blue, 2: orange, 3: green,
> 4: red}. In the input image, the class 1 and 2 are in red while the class
> 3 and 4 are in green.
> *Categories Predicted by the Deterministic Model*:
> The deterministic model was trained to predict categories based on the
> colors of the rectangles. However, due to the inherent randomness in our
> dataset, where each color can correspond to two different categories, the
> deterministic model tends to predict only one of the two possible classes,
> namely classes 1 and 4 as seen in Figure 2b.
>
> > Regarding the difference with [1]:
>
> We were unaware of [1] and apologize
> for failing to cite the other relevant related papers. We will include these
> in the related works and the LIDC results table when relevant. The work in
> [1] is indeed very close to ours. They also use a discrete diffusion model,
> which they name categorical, and add image conditioning in a similar
> way. Our methods are very similar, however the two papers overall have
> multiple differences: (1) we additionally propose an autoregressive
> scheme for doing future prediction. In this case, our network is performing
> ‘segmentation to segmentation’, which means the input to the network
> is only one or multiple previous segmentation maps. (2) we provide a
> motivating example that clearly shows the sampling ability of the model
> and in particular the fact that it properly follows the data distribution.
> (3) we evaluate on different datasets: LIDC aside, we introduce a new car
> intersection simulation dataset designed to offer a finite set of valid future
>  trajectories; we evaluate on the future prediction task on the Cityscapes Dataset.
>
> > Regarding the applicability of Chen et al. (2022a):
>
> Their approach would also be applicable but is significantly
> different. First, their way of handling the discrete aspect of segmentation
> is different. In fact, they do not use discrete diffusion, they are instead
> mapping discrete classes to a continuous vector representation and then
> doing continuous diffusion. Also, note that their work focuses on panoptic
> segmentation, which might necessitate modifications to the network structure. Nonetheless, leveraging instance data from datasets where it exists
> could prove advantageous.
>
> > For each car in each validation example 10 samples
>  are generated. The best of the 10 samples is selected according to
> FDE. Then the
> mean of the best FDE values (hit and miss) is computed. Is this correct?
> I am not sure the mean is very informative as the distribution seems to be quite skewed.
>
> You are correct, we select the best FDE of the ten
> samples and then perform the mean over the validation set. The mean is
> informative, but you are correct in saying that a high miss can influence
> it strongly, which is why we also report the miss rate and consider it as
> our main ranking measure, and we will emphasize it more in the updated
> paper.
>
> > Regarding the Cityscapes experiment:
>
> Thank you for this relevant question.
> We noticed that there is not as much ambiguity in the Cityscape's future
> segmentation task as one could expect. Indeed, given 3 past frames,
> the long-term future (1.5s) has little space for unexpected events, and we
> should note that Cityscapes does not feature a very chaotic environment,
> with disciplined German pedestrians. Therefore, a deterministic model
> can already do a very good job at predicting future segmentation, and
> the differences will mostly lie in slight speed variations, which will result
> in slight displacements of object boundaries. Multiple sampling allows to
> cover more of these possible speed variations, but do not generate completely
>  different scenarios, as in the car intersection simulation. While we
> would ideally be able to answer a question such as ”will the child cross the
> road”, the nature of Cityscapes and the relatively short time delta of the prediction with the current frame does not allow us to showcase such things happening, but we hope our work is a step in that direction.

---

> > ### Comment · Reviewer_qH1j · 2023-11-22
> >
> > Thank you for your replies to my comments and the comments of the other reviewers.
> > Overall most concerns have been addressed sufficiently, and I update my score respectively.
> >
> > Some more notes:
> > * If there is not a lot of ambiguity in the CityScapes dataset, maybe it is not a good choice for this paper?
> > * I am still not sure what a mean FDE of 68 or 11.9 is supposed to tell me. Yes, higher is worse, but beyond that? Even for the deterministic model with a mean of 68 the median is still ~2. In addition, why is it better to choose a wrong exit that is closer than one that is further away? (wrt to the question how well the ambiguity is modeled) Which wrong exit is chosen has a strong impact on the mean.
> > * Wrt Chen et al. (2022a), panoptic segmentation is in the end just semantic+instance, not using the instance part should be trivial. It would still be interesting if there are pros and cons to your approach vs their approach or if there is no difference.

---

### Meta-Review · Area_Chair_SmFA · 2023-12-06

**Metareview:**

In this paper, the authors propose to use discrete diffusion models to model uncertainty in semantic segmentation, which enables segmenting ambiguous context. The main contributions include the adaptation of discrete diffusion to model uncertainty and the introduction of an auto-regressive diffusion framework for future forecasting. The authors have validated experiments on some synthetic and real datasets.

The main concerns raised by the reviewers include poor experiments (missing baselines on LIDC dataset), missing experiments on uncertainty estimation. In the rebuttal, authors claim that uncertainty estimation is not the focus of the work, and it is merely used as a tool. However, many of the reviewers got confused by this and were expecting results on uncertainty estimation. I would encourage the authors to work on this and restructure the narrative to make this point more clear.

Even though uncertainty estimation is not the focus, since the authors use it as a component in their method, evaluating this might be interesting. It would have been nice if the authors included at least one table of results on this.

Since the goal of the paper is primarily semantic segmentation, I would expect to see extensive evaluation on some large-scale datasets like Cityscapes. While there is one table in the paper, as pointed by reviewer qH1j, more detailed analysis is missing. Without such analysis, I feel the paper is not in a state of acceptance right now.

So, I would encourage the authors to include more experiments, extensive analysis, show some results on uncertainty estimation and restricting the narrative. As such, the paper is not in state to publish right now and I recommend rejecting this paper.

**Justification For Why Not Higher Score:**

The paper needs change in narrative, more quantitative analysis, and some experiments on uncertainty estimation. The paper, in the current form, is not in a state of publishing.

**Justification For Why Not Lower Score:**

N/A

---

### Decision · Program_Chairs · 2024-01-16

Reject